# In Situ Growth of Ti$_3$C$_2$/UiO-66-NH$_2$ Composites for Photoreduction of Cr(VI)

Huan He [1], Xusheng Wang [1,2,*], Qin Yu [1], Wenbin Wu [1], Xinya Feng [1], Deyu Kong [1], Xiaohui Ren [3] and Junkuo Gao [1,*]

1   Institute of Functional Porous Materials, School of Materials Science and Engineering, Zhejiang Sci-Tech University, Hangzhou 310018, China; 202030302112@mails.zstu.edu.cn (H.H.); 2021316101103@mails.zstu.edu.cn (Q.Y.); 2022316101108@mails.zstu.edu.cn (W.W.); 2022316101030@mails.zstu.edu.cn (X.F.); 2022316101070@mails.zstu.edu.cn (D.K.)
2   Guangdong Provincial Key Laboratory of Functional Supramolecular Coordination Materials and Applications, College of Chemistry and Materials Science, Jinan University, Guangzhou 510632, China
3   The State Key Laboratory of Refractories and Metallurgy, School of Materials and Metallurgy, Wuhan University of Science and Technology, Wuhan 430081, China; xhren@wust.edu.cn
*   Correspondence: xswang@zstu.edu.cn (X.W.); jkgao@zstu.edu.cn (J.G.)

**Abstract:** Cr(VI) is one of the most toxic heavy metals, posing multiple threats to humans and ecosystems. Photoreduction of toxic Cr(VI) to para-toxic Cr(III) is one of the most effective ways to remove heavy metal chromium but is still challenging. Herein, Ti$_3$C$_2$/UiO-66-NH$_2$ composites with different ratio of Ti$_3$C$_2$ were synthesized via an in situ solvothermal process and used for the enhanced photocatalytic removal of Cr(VI) for the first time. The UiO-66-NH$_2$ nanoparticles were dispersed on the surface and slits of accordion-like Ti$_3$C$_2$ homogeneously. A strong interfacial interaction between Ti$_3$C$_2$ and UiO-66-NH$_2$ was formed, which was indicated by the XPS. The Fermi level of the MXene cocatalyst is lower than UiO-66-NH$_2$; thus, Ti$_3$C$_2$ can serve as the electron sink and accumulate photogenerated electrons from UiO-66-NH$_2$ on its surface. A depletion layer was also formed due to the different Fermi levels of UiO-66-NH$_2$ and Ti$_3$C$_2$, which prevents electrons from flowing back to UiO-66-NH. The strong interfacial interaction and formed depletion layer are beneficial for the following charge transfer from UiO-66-NH$_2$ to Ti$_3$C$_2$ after light irradiation and for suppressing the photogenerated charge recombination. With suitable band positions and enhanced charge separation ability, Ti$_3$C$_2$/UiO-66-NH$_2$ composites exhibited better photoreduction efficiency of Cr$_2$O$_7^{2-}$ than pure Ti$_3$C$_2$ and UiO-66-NH$_2$, with optimized samples reaching 100% in 40 min. The photoreduction kinetics of Cr$_2$O$_7^{2-}$ by 2-T/U was also studied, with a photoreduction rate of 0.0871 min$^{-1}$, which is about 2.6 times higher than that of the pure UiO-66-NH. This research provides a new type of efficient and environmentally friendly photocatalysts for the photoreduction of Cr$_2$O$_7^{2-}$.

**Keywords:** metal–organic frameworks; photocatalysis; MXene; Cr(VI); photocatalyst; photoreduction; porous materials

## 1. Introduction

The concentration of heavy metals (chromium, lead, arsenic, cadmium, mercury, etc.) in aquatic systems has increased at an alarming rate with the development of the economy. These heavy metals have posed multiple threats to humans and ecosystems. Of the various heavy metals, chromium is reported to be one of the most common and toxic contaminants in aquatic systems and is in the list of top 20 hazardous pollutants [1]. Generally, Cr(VI) is toxic, while Cr(III) is para-toxic and easily forms a precipitate that can be removed [2]. The main sources of Cr(VI) are diverse, including the metal processing, dyes, automobile, leather, mining, and textile industries. Thus, it is important to develop effective purification

technologies including chemical precipitation, adsorption, filtration, and biological treatment. However, the use of these techniques also suffers from expensive processing, high consumption of chemicals, and poor possibilities of sustainable development [3]. Therefore, it is urgent to find an effective way to eliminate emerging hazardous pollutants such as Cr(VI). Photocatalytic techniques are an emerging method for solar fuel production and environmental remediation, with the help of clean solar energy and high performance photocatalysts [4–17]. In this respect, photocatalytic techniques are also found to be an effective and ecofriendly approach to convert toxic Cr(VI) into hypo-toxic Cr(III) forms [18–21].

Metal–organic frameworks (MOFs) are a kind of emerging porous material with a highly ordered network, high adsorption capacity, tunable pore sizes, large surface area, and rich active sites [22–25]. These outstanding characteristics have made them successfully applicable in gas capture/storage/separation, photo/electro-catalysis, sensing, drug delivery, batteries, etc. [26–31]. Constructing MOFs with light harvesting abilities is easy and convenient because of the richness of organic linkers, thereby making them unique platforms for photocatalysis [32–35]. In recent years, several MOFs have been utilized for hexavalent chromium reduction in aqueous solutions [2,20,21,36]. For example, two famous MOFs, UiO-66-$NH_2$ and MIL-125-$NH_2$, were first applied for this reaction, due to their light absorption ability and porous structures [37,38]. Further functionalization of MOFs by introducing strong anions adsorption structures and visible-light absorption porphyrin units simultaneously, allowing for MOFs with stronger Cr(VI) ions adsorption and light-harvesting abilities, and further enhancing the photoreduction efficiency [39]. Though some progress has been achieved using pure MOFs for photoreduction of toxic Cr(VI) to para-toxic Cr(III), the lack of enough active sites and the low conductivity of MOFs has still restricted their catalytic performance.

MXenes are a kind of two-dimensional (2D) material, including transition metal carbides, nitrides, or carbonitrides [40]. They have excellent mechanical, electrical, and thermal properties, and can be applied for photo/electro-catalysis, energy storage, sensors, etc. [41]. They can also be further functionalized to expand their range of applications and properties [42]. With the excellent electrical conductivity and abundant uncoordinated metal sites of $Ti_3C_2$ MXene, intimate contact with MOFs has been achieved, benefitting the further separation of photogenerated charge carriers during photocatalysis [43,44]. Thus, formed functional photocatalysts combining MXene with MOFs are expected to be suitable for the photoreduction of toxic Cr(VI) to para-toxic Cr(III), but have not yet been realized.

Herein, $Ti_3C_2$/UiO-66-$NH_2$ with different ratio of $Ti_3C_2$ was synthesized via an in situ solvothermal process for enhancing the photoreduction efficiency of Cr(VI) for the first time. The UiO-66-$NH_2$ nanoparticles were dispersed on the surface and slits of accordion-like $Ti_3C_2$ homogeneously. A strong interfacial interaction between $Ti_3C_2$ and UiO-66-$NH_2$ was formed, as indicated by the XPS. The Fermi level of the MXene cocatalyst is lower than UiO-66-$NH_2$; thus, $Ti_3C_2$ can work as an electron sink, accumulating photogenerated electrons from UiO-66-$NH_2$ on its surface. A depletion layer was also formed due to the different Fermi levels of UiO-66-$NH_2$ and $Ti_3C_2$, which prevents electrons from flowing back to UiO-66-NH. The strong interfacial interaction and formed depletion layer are beneficial for the following charge transfer from UiO-66-$NH_2$ to $Ti_3C_2$ after light irradiation and for suppressing the photogenerated charge recombination. With suitable band positions and enhanced charge separation ability, $Ti_3C_2$/UiO-66-$NH_2$ composites exhibited better photoreduction efficiency of $Cr_2O_7^{2-}$ than pure $Ti_3C_2$ and UiO-66-$NH_2$, with optimized samples reaching 100% in 40 min. The photoreduction mechanism was also well studied.

## 2. Results and Discussion

### 2.1. Structural Characterizations

A schematic diagram for $Ti_3C_2$ MXene and $Ti_3C_2$/UiO-66-$NH_2$ composites synthesis is shown in Scheme 1. The commercial $Ti_3AlC_2$ powder showed typical peaks of $Ti_3AlC_2$ phase based on JCPDS card #52-0875 centered at 9.62° (002), 19.24° (004), 34.10° (101), 36.82° (103), 39.08° (104), 41.84° (105), and 48.53° (107) (Figure 1a). $Ti_3C_2$ MXene was synthesized

by etching $Ti_3AlC_2$ powder with HF [45]. The diffraction peak at 39.08° (104) in $Ti_3AlC_2$ disappeared after etching, which demonstrated the successful elimination of Al layers after HF etching to form 2D $Ti_3C_2$ MXene. The main peaks of the obtained $Ti_3C_2$ MXene centered at 9.0° (002) and 18.3° (004) were slightly shifted from the 9.62° (002) and 19.24° (004) of $Ti_3AlC_2$, indicating a change in c-lattice parameter after elimination of Al layers [46]. The obtained $Ti_3C_2$ MXene also showed similar peaks to those in previous reported literature (9.0°, 18.3°, 27.7°, 36.1°, 41.9°, and 61.0°) [47]. $UiO-66-NH_2$ was further prepared using the traditional solvothermal method [48]. The successful synthesis of $UiO-66-NH_2$ was also proved by PXRD patterns with peak positions at 7.36°, 8.48°, 12.04°, 25.68°, and 33.12° in the region of 3–50°, which is similar to the previous reported literature (Figure 1b) [49]. $Ti_3C_2/UiO-66-NH_2$ composites were further synthesized by in situ growth of MOFs on the $Ti_3C_2$ surface in solvothermal conditions. Contents of 5 wt%, 10 wt%, 15 wt%, and 20 wt% $Ti_3C_2$ in $Ti_3C_2/UiO-66-NH_2$ composites were obtained by adjusting the ratio of $Ti_3C_2$ and the precursor $UiO-66-NH_2$, and named as 1-T/U, 2-T/U, 3-T/U, and 4-T/U, respectively. The 1-T/U and 2-T/U had no obvious characteristic peak of $Ti_3C_2$ (9°), which is attributed to their low content of MXene (Figure 1b). As expected, the characteristic peak of $Ti_3C_2$ (9°) appeared in 3-T/U and 4-T/U due to the increasing content of $Ti_3C_2$.

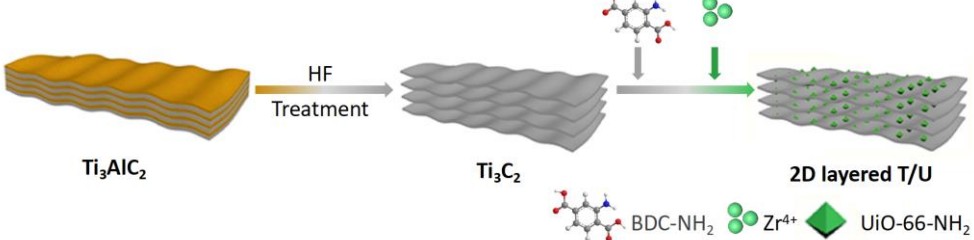

**Scheme 1.** Schematic diagram for $Ti_3C_2$ MXene and $Ti_3C_2/UiO-66-NH_2$ composites synthesis.

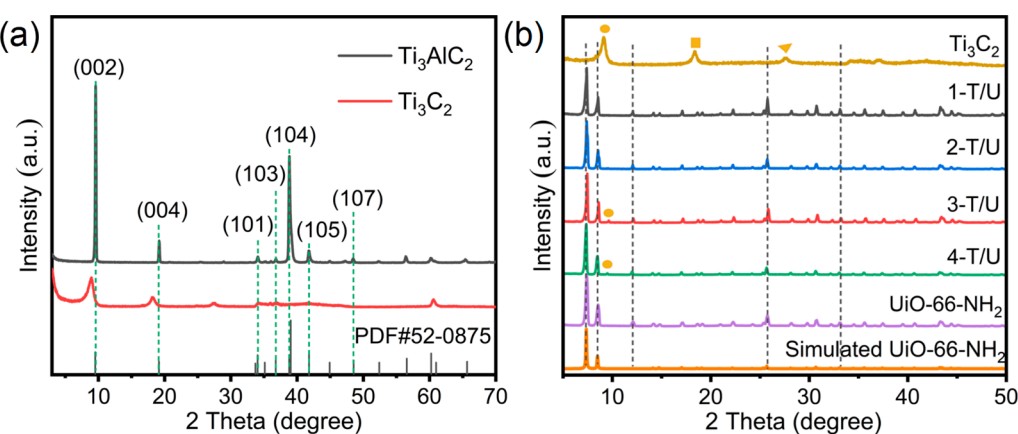

**Figure 1.** (**a**) PXRD patterns of $Ti_3C_2$ and $Ti_3AlC_2$; (**b**) PXRD patterns of $Ti_3C_2$, $UiO-66-NH_2$, 1-T/U, 2-T/U, 3-T/U and 4-T/U.

The distribution of $UiO-66-NH_2$ in the $Ti_3C_2/UiO-66-NH_2$ composites was characterized by scanning electron microscopy (SEM). $Ti_3C_2$ exhibited accordion-like structures after etching by HF, further confirming that the Al layers of $Ti_3AlC_2$ were successfully removed (Figure 2a). The morphology of $UiO-66-NH_2$ is regularly octahedral, which was also revealed by the SEM pictures (Figure 2b). The morphology of both $UiO-66-NH_2$ and $Ti_3C_2$ were maintained in the $Ti_3C_2/UiO-66-NH_2$ composites, and the $UiO-66-NH_2$ nanoparticles were dispersed on the surface and slits of $Ti_3C_2$ (Figure 2c–f). EDS and mapping of $Ti_3C_2/UiO-66-NH_2$ composites has been added as Figures S1 and S2 and Table S1. Take 4-T/U as an example: all the elements of O, N, Ti, F, and Zr can be found in the EDX of 4-T/U (Figure S1, Table S1). A low Ti element content was found, with 1.27 wt.% and 0.52 atom%, respectively, indicating that $Ti_3C_2$ had been well covered by UiO-66-NH.

The mapping of $Ti_3C_2$/UiO-66-$NH_2$ composites further indicated that the UiO-66-$NH_2$ nanoparticles were dispersed homogeneously on the surface and slits of $Ti_3C_2$ (Figure S2).

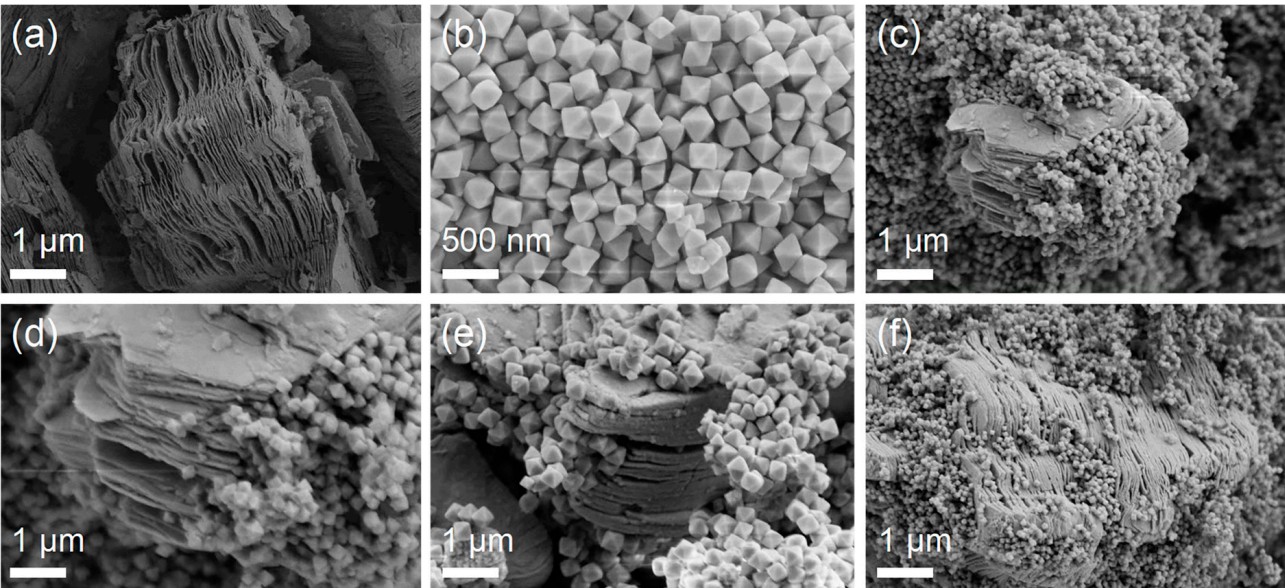

**Figure 2.** SEM images of $Ti_3C_2$ (**a**), UiO-66-$NH_2$ (**b**), 1-T/U (**c**), 2-T/U (**d**), 3-T/U (**e**), and 4-T/U (**f**).

The valence state, chemical composition, and possible interfacial interaction between UiO-66-$NH_2$ and MXene of $Ti_3C_2$/UiO-66-$NH_2$ composites were further characterized by X-ray photoelectron spectroscopy (XPS) (Figures 3 and S3). All the spectra were calibrated by C 1s (Figure S3). Figure 3a showed the survey spectra of $Ti_3C_2$, 2-T/U, and UiO-66-NHIn $Ti_3C_2$. F and O elements were present but not C and Ti, indicating surface functional groups were introduced after etching the $Ti_3AlC_2$ powder. All the Zr, C, O, and N elements were present in UiO-66-$NH_2$, and both $Ti_3C_2$ and UiO-66-$NH_2$ were proven to be present in 2-T/U because of the existence of all elements of those two parts. High-resolution Ti 2p XPS in $Ti_3C_2$ and 2-T/U were further studied in Figure 3b. Ti 2p in $Ti_3C_2$ can be deconvoluted into two pairs of peaks with Ti $2p_{3/2}$, located at 455.2 and 456.4 eV, which were related with Ti-C and terminal Ti-O/F, respectively (Figure 3b) [50]. Compared with $Ti_3C_2$, Ti-C in 2-T/U moved to 455.0 eV and terminal Ti-O/F moved in the opposite direction to 458.0 eV (Figure 3b). The shifting of the binding energy of Ti 2p in 2-T/U might be ascribed to the newly formed coordination bond between carboxylate and Ti from UiO-66-$NH_2$ and $Ti_3C_2$, respectively. Therefore, a strong interfacial interaction between $Ti_3C_2$ and UiO-66-$NH_2$ in 2-T/U was formed, which is beneficial for the following charge transfer between UiO-66-$NH_2$ and $Ti_3C_2$ after light irradiation. In addition, compared with UiO-66-$NH_2$, the binding energy of both N 1s and Zr 3d in 2-T/U were not moved in 2-T/U, implying those two elements were not participants in the formation of a new coordination bond (Figure 3c,d).

FT-IR spectroscopy analysis of the chemical structures of the as-prepared samples was also conducted. As shown in Figure 4a, the FT-IR spectra of x-T/U (x = 1, 2, 3, 4) are very similar to those of UiO-66-$NH_2$, indicating the presence of UiO-66-$NH_2$ in the x-T/U photocatalyst. The large and broad absorption peak at 3430 $cm^{-1}$ is due to the -$NH_2$ and -OH groups in UiO-66-$NH_2$ [51]. The two peaks at 1580 $cm^{-1}$ and 1440 $cm^{-1}$ are attributed to the stretching vibrations of the benzene ring, with a carbon-based stretching vibration also observed at 1580 $cm^{-1}$ [52]. The peaks at 1260 $cm^{-1}$ and 1390 $cm^{-1}$ were assigned to the C-N bond, while the peak at 769 $cm^{-1}$ is likely because of the C-H bond bending vibration in the benzene ring [53]. The peak at 667 $cm^{-1}$ was assigned to the Zr-O bond stretching vibration [54].

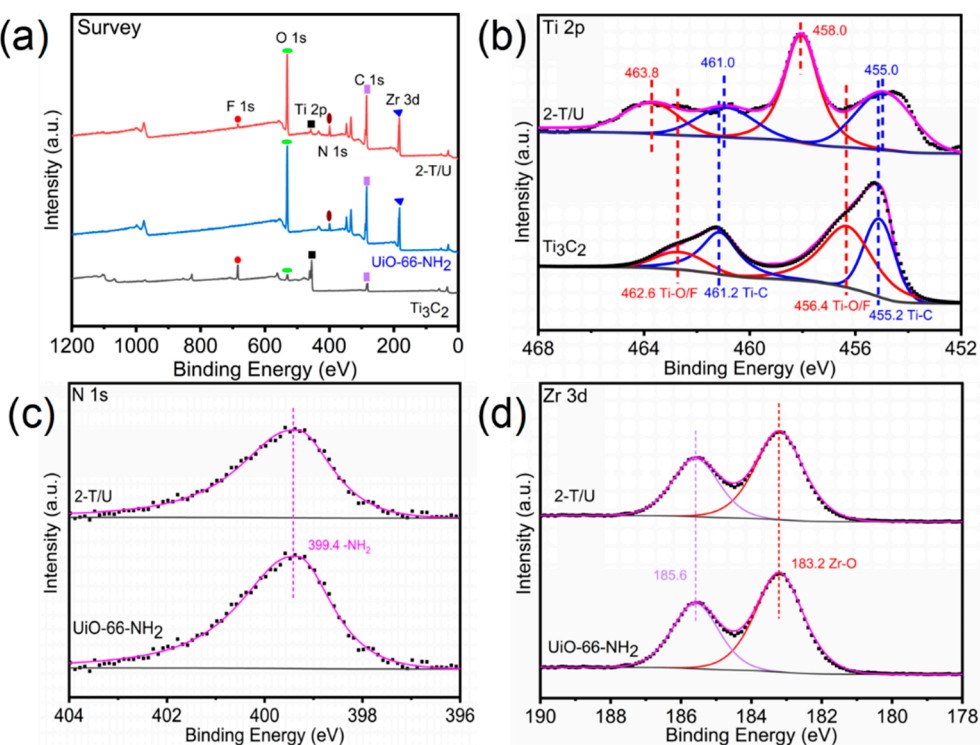

**Figure 3.** (**a**) X-ray photoelectron survey spectra of $Ti_3C_2$, UiO-66-$NH_2$ and 2-T/U; high-resolution X-ray photoelectron spectra of Ti 2p (**b**) in $Ti_3C_2$ and 2-T/U, and N 1s (**c**) and Zr 3d (**d**) in 2-T/U and UiO-66-$NH_2$. Blue and red lines in (**b**) are corresponding to Ti-C and Ti-O/F, respectively. Red and purple lines in (**d**) are corresponding to Zr $3d_{5/2}$ and Zr $3d_{3/2}$ Zr-O, respectively.

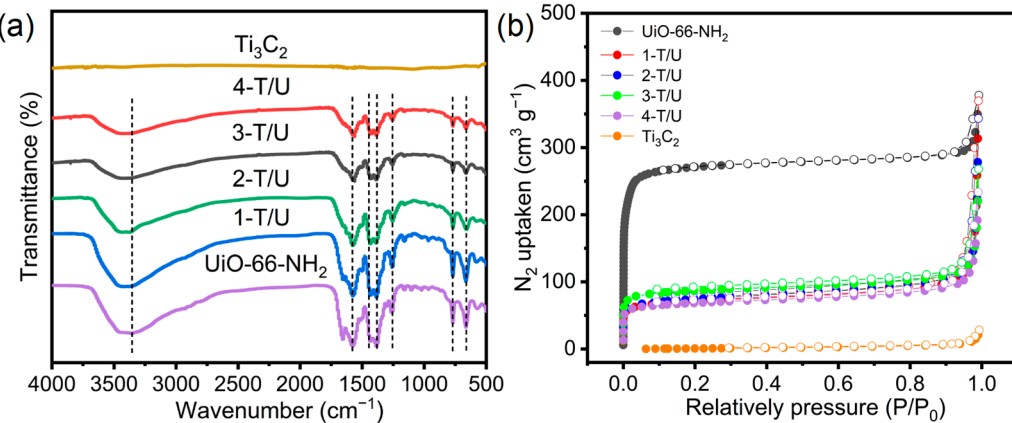

**Figure 4.** The FTIR spectra (**a**) and $N_2$ sorption isotherms (**b**) of $Ti_3C_2$, UiO-66-$NH_2$ and their composites.

The porosity properties of the photocatalysts were measured using $N_2$ sorption isotherms (Figure 4b). UiO-66-$NH_2$ showed type-I isotherms with high $N_2$ sorption levels, suggesting it is a highly porous material with the presence of uniform micropore. After combination with $Ti_3C_2$, the $Ti_3C_2$/UiO-66-$NH_2$ composites still kept their highly porous structure. The highly porous structure of photocatalysts will benefit their concentration of pollution and light harvesting abilities [55].

## 2.2. Photoelectrochemical Characterizations of Photocatalysts

To study the light harvesting abilities and bandgap of the prepared samples, solid UV–Vis diffusion reflectance spectroscopy (UV–Vis DRS) spectra of the photocatalysts were further investigated. $Ti_3C_2$ showed a high visible light absorption range but had no obvious absorption edge due to its metallic nature (Figure 5a) [50,55]. UiO-66-$NH_2$

exhibited a strong visible light harvesting ability to 450 nm. Compared with UiO-66-NH$_2$, a red shift of the absorption range was found in 2-T/U. The bandgap energy (Eg) of UiO-66-NH$_2$ and 2-T/U were further calculated using the Kubelka–Munk equation $((\alpha h\nu)^2 = A(h\nu - Eg))$. As shown in Figure 5b, the bandgap of 2-T/U was calculated to be 2.86 eV, which was somewhat smaller than that of the pure UiO-66-NH$_2$ (2.93 eV). The decreasing of the bandgap originated from the formed heterojunction between UiO-66-NH$_2$ and Ti$_3$C$_2$, which affected the later photoreduction performance [56]. To fully obtain the bandgap information of UiO-66-NH$_2$, the conduction band (CB) potential was investigated by Mott–Schottky plot. As shown in Figure 6a, UiO-66-NH$_2$ showed a typical n-type semiconductor behavior with positive slopes. The CB potential of UiO-66-NH$_2$ was calculated to be $-0.79$ eV versus NHE. Combined with the bandgap of 2.93 eV, the valence band (VB) can be calculated to be 2.14 eV. In Ti$_3$C$_2$/UiO-66-NH$_2$, the Fermi level of MXene cocatalyst is lower than UiO-66-NH. Therefore, Ti$_3$C$_2$ will serve as the electron sink, and lower the CB position of Ti$_3$C$_2$/UiO-66-NH. With the VB of 2.14 eV and bandgap of 2.86 eV of 2-T/U, the CB position can be calculated to be $-0.72$ eV.

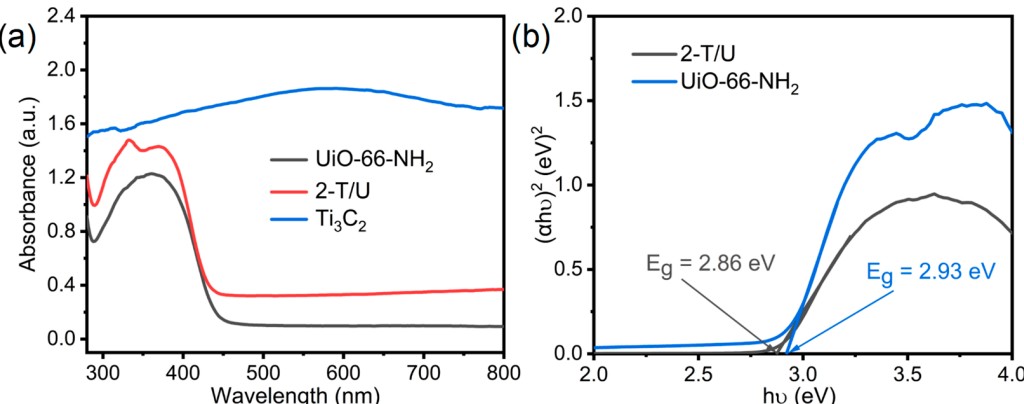

**Figure 5.** (**a**) Solid UV–Vis diffusion reflectance spectroscopy (DRS) spectra of Ti$_3$C$_2$, 2-T/U and UiO-66-NH$_2$; (**b**) plots derived from UV–Vis DRS spectra of 2-T/U and UiO-66-NH$_2$ for bandgap calculation.

To explore the photogenerated charge separation ability, transient photocurrent (TPC) response and steady photoluminescence (PL) spectra were studied. The TPC responses of UiO-66-NH$_2$ and 2-T/U were conducted under visible light irradiation. As shown in Figure 6b, the 2-T/U composite showed a much stronger signal compared with UiO-66-NH$_2$, indicating a more efficient photogenerated charge separation ability. The excellent performance of 2-T/U might be attributed to the Schottky heterojunction formed between UiO-66-NH$_2$ and Ti$_3$C. The excellent charge separation ability of 2-T/U was also confirmed by the steady PL spectra, where the excitation wavelength was set at 390 nm. As shown in Figure 6c, UiO-66-NH$_2$ showed a strong emission peak at 465 nm, which can be attributed to the high recombination rate of photo-generated charge carriers. After the introduction of Ti$_3$C$_2$, the peak intensity of 2-T/U decreased significantly, indicating that the recombination of the charge carriers was greatly suppressed [57]. The charge transfer efficiency of UiO-66-NH$_2$ and 2-T/U were also studied using electrochemical impedance spectroscopy (EIS). As shown in Figure 6d, 2-T/U showed the smaller arc radius compared with UiO-66-NH$_2$, indicating a better charge transfer ability [43].

### 2.3. Photocatalytic Performance of Photocatalysts

Encouraged by the strong light harvesting ability and efficient separation ability of photogenerated charge carriers, the Ti$_3$C$_2$/UiO-66-NH$_2$ composites were applied for the photoreduction of toxic Cr(VI) to para-toxic Cr(III) under visible light irradiation. Previous studies have revealed that the photoreduction of Cr(VI) can be accelerated at low pH values, and in acidic conditions the present form of Cr(VI) is mainly Cr$_2$O$_7^{2-}$ anions [39]. Therefore, this reaction was conducted at pH = 2, and the residual Cr$_2$O$_7^{2-}$ was monitored by the

UV–Vis spectra of diphenylcarbazide (DPC)-$Cr_2O_7^{2-}$ solutions. As shown in Figure 7a,b, MXene can only adsorb $Cr_2O_7^{2-}$ but not convert it to $Cr^{3+}$ under visible light. In contrast, both UiO-66-$NH_2$ and $Ti_3C_2$/UiO-66-$NH_2$ composites can promote the adsorption of $Cr_2O_7^{2-}$ anions, and further convert the adsorbed $Cr_2O_7^{2-}$ to reduced $Cr^{3+}$ smoothly. The removal of $Cr_2O_7^{2-}$ by UiO-66-$NH_2$ can reach 80% after 40 min visible light irradiation. The removal was enhanced dramatically after the introduction of MXene in the $Ti_3C_2$/UiO-66-$NH_2$ composites, especially for the optimal 2-T/U, which can reach 100% under the same conditions. Digital photographs of the solution also prove that almost of all the $Cr_2O_7^{2-}$ anions had been removed (Figure S4). The effect of catalyst dose and substrate concentration were also examined (Figure S5). A higher dose of $Ti_3C_2$/UiO-66-$NH_2$ is beneficial for the photoreduction rate, and it is hard to remove all the $Cr_2O_7^{2-}$ anions with a concentration above 100 ppm.

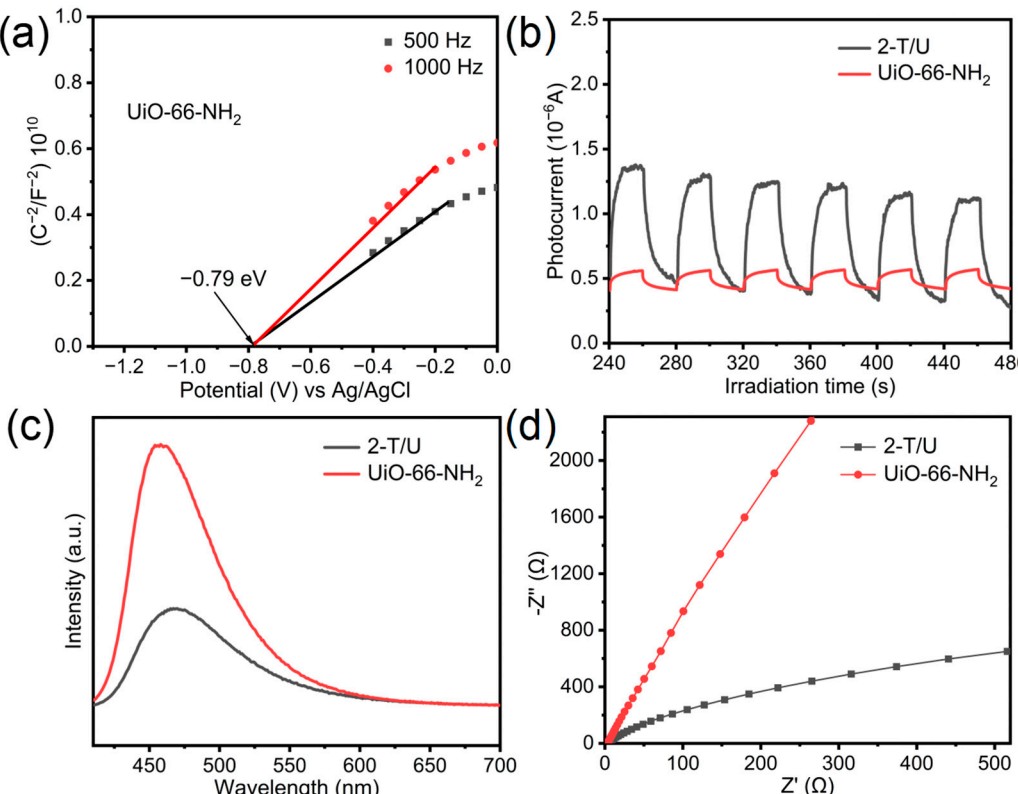

**Figure 6.** (**a**) Mott–Schottky plots for sample of UiO-66-$NH_2$; (**b**) transient photocurrent response, (**c**) steady photoluminescence spectra, and (**d**) electrochemical impedance spectroscopy for samples of UiO-66-$NH_2$ and 2-T/U.

To further evaluate their photoreduction performance, the photoreduction kinetics (Figure 7c) of $Cr_2O_7^{2-}$ by photocatalysts were examined based on the data in Figure 7a and all the samples were found to follow pseudo-first-order kinetics. The observed photoreduction rate of the optimized $Ti_3C_2$/UiO-66-$NH_2$ composite, 2-T/U, achieved a photoreduction rate of 0.0871 $min^{-1}$, which is about 2.6 times higher than that of the pure UiO-66-NH. The much higher photoreduction rate of 2-T/U might be ascribed to the formed heterojunction structure between $Ti_3C_2$ and UiO-66-$NH_2$, which can promote the separation of photogenerated electrons and holes [43]. The comparison of photoreduction performance of our results with previous studies also proved that 2-T/U is an excellent photocatalyst for the removal of $Cr_2O_7^{2-}$ anion (Tables S2–S4).

Moreover, 2-T/U showed excellent chemical stability, which had been proved by the recycling experiments, and retained PXRD patterns after photoreduction (Figures 7d and S6).

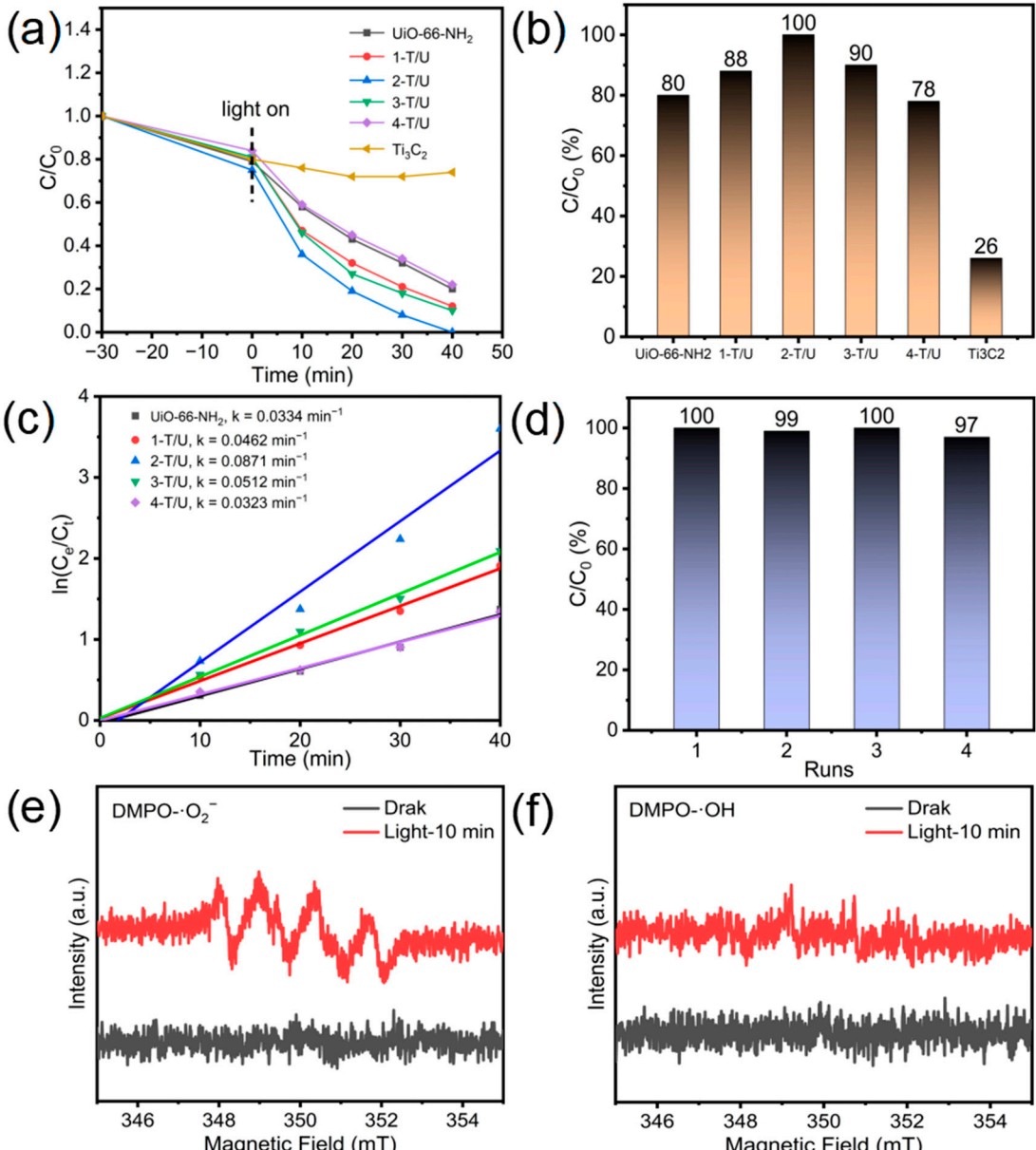

**Figure 7.** (**a**,**b**) Removal of $Cr_2O_7^{2-}$ over $Ti_3C_2$, UiO-66-NH$_2$, 1-T/U, 2-T/U, 3-T/U, and 4-T/U; (**c**) photoreduction kinetics and rate of $Cr_2O_7^{2-}$ over photocatalysts (UiO-66-NH$_2$, 1-T/U, 2-T/U, 3-T/U, and 4-T/U); (**d**) recycling ability of 2-T/U; the ESR spectra of active species trapped by DMPO in aqueous dispersion for $\cdot O_2^-$ (**e**) and $\cdot OH$ (**f**) over 2-T/U.

ESR analyses of $\cdot O_2^-$ and $\cdot OH$ produced by $Ti_3C_2$/UiO-66-NH$_2$ were further conducted, as shown as Figure 7e,f. These species were not detected without light irradiation. After exposure to visible light for 10 min, obvious DMPO-$\cdot O_2^-$ and DMPO-$\cdot OH^-$ peaks were detected [58]. The photoreduction mechanisms of $Cr_2O_7^{2-}$ by $Ti_3C_2$/UiO-66-NH$_2$ were further proposed based on the previous results. In $Ti_3C_2$/UiO-66-NH$_2$, the Fermi level of the MXene cocatalyst is lower than that of UiO-66-NH. Therefore, $Ti_3C_2$ can serve as the electron sink, and accumulates photogenerated electrons from UiO-66-NH$_2$ on its surface [59]. A depletion layer was also formed due to the different Fermi levels of UiO-66-NH$_2$ and $Ti_3C_2$, which prevents electrons from flowing back to UiO-66-NH. Therefore, the accumulated electrons on $Ti_3C_2$ in the $Ti_3C_2$/UiO-66-NH$_2$ composites can further reduce the $Cr_2O_7^{2-}$ to $Cr^{3+}$. These accumulated electrons can also reduce $O_2$ to $\cdot O_2^-$ species, which was detected using ESR experiments. The produced $\cdot O_2^-$ species can also help

to reduce the $Cr_2O_7^{2-}$ to $Cr^{3+}$ [60]. Correspondingly, the residual holes in the VB of $Ti_3C_2$/UiO-66-NH$_2$ can oxide $H_2O$ to ·OH, which was proved via ESR experiments. The formed ·OH mainly facilitates the separation of photogenerated charge carriers to improve the reduction of $Cr_2O_7^{2-}$ [60].

### 3. Materials and Methods

*3.1. Chemical Regents*

Titanium aluminum carbide ($Ti_3AlC_2$, 98%, 200 mesh, Maclin), zirconium chloride ($ZrCl_4$, 98%, Maclin), 2-aminoterephthalic acid (BDC-NH$_2$, 98%, Maclin), and hydrofluoric acid (HF, 40%, Maclin), N, N-dimethylformamide (AR, DMF, Aladdin), acetic acid (AR, HAc, Aladdin), methanol (AR, Aladdin), absolute ethyl alcohol (AR, Aladdin), $K_2Cr_2O_7$ (99.8%, Tianjin Jinnan), isopropanol (AR, Aladdin), $H_2SO_4$ (AR, 95–98%, Shuanglin chemical regent), $Na_2SO_4$ anhydrase (AR, Maclin), Nafion 'D-521 dispersion, 5% w/w in water and 1-propanol (Alfa Aesar), diphenylcarbazide (AR, Maclin), acetone (99.5%, Shuanglin chemical regent), and HCl (36–38%, Shuanglin chemical regent) were used directly without further purification.

*3.2. Synthesis of Accordion-Like $Ti_3C_2$ MXene*

$Ti_3C_2$ MXenes were synthesized based on the previous literature [1]. Typically, 1 g $Ti_3AlC_2$ powder was mixed with 35 mL of 40% concentrated HF while stirring at room temperature for 24 h. Then, the suspension was ultrasonically treated for 1 h in an ultrasonic bath. The obtained suspension was washed with deionized water and alcohol by centrifugation until the pH of the solution reached 2. At last, the $Ti_3C_2$ MXene powder was obtained after freeze-drying for 2 days.

*3.3. Synthesis of UiO-66-NH$_2$*

UiO-66-NH$_2$ was synthesized based on the previous reported literature with a slight modification [48]. A typical synthetic procedure was used, as follows: 0.1433 g $ZrCl_4$ was dissolved in 25 mL DMF for 1 h stirring, then 0.1114 g BDC-NH$_2$ was added with another 20 min stirring. Afterwards, 5 mL of HAc was added to the above solution, and then transferred into a 100 mL autoclave heated at 120 °C for 4 h. After cooling naturally to room temperature, the precipitate was washed with DMF and anhydrous methanol. Finally, the obtained yellow solids were dried at 80 °C in a vacuum overnight to obtain the UiO-66-NH$_2$.

*3.4. Synthesis of $Ti_3C_2$/UiO-66-NH$_2$*

The $Ti_3C_2$/UiO-66-NH$_2$ was synthesized as follows: 0.1433 g $ZrCl_4$ was pre-dissolved in a solvent mixture of 25 mL DMF and 5 mL HAc, then quantitative $Ti_3C_2$ (25, 50, 75, and 100 mg) were added into the above solution. After stirring for 1 h, pre-dissolved BDC-NH$_2$ (0.137 g) in 10 mL DMF was added into the solution and stirred for another 0.5 h. Afterwards, the above mixture was transferred into a 100 mL autoclave and kept at 120 °C for 24 h. After cooling naturally to room temperature, the precipitate was washed with DMF and anhydrous methanol. Finally, the obtained yellow solids were dried at 80 °C in a vacuum overnight, and named as 5 wt% $Ti_3C_2$/UiO-66-NH$_2$, 10 wt% $Ti_3C_2$/UiO-66-NH$_2$, 15 wt% $Ti_3C_2$/UiO-66-NH$_2$, and 20 wt% $Ti_3C_2$/UiO-66-NH$_2$, abbreviated as 1-T/U, 2-T/U, 3-T/U, and 4-T/U, respectively.

*3.5. Material Characterizations*

The purity and crystal phase of the photocatalysts were characterized using a powder X-ray diffractometer (PXRD, D8 ADVANCE, Bruker, MA, USA). The specific surface areas (Brunauer−Emmett−Teller, BET) of those photocatalysts were calculated using $N_2$ sorption isotherms measured at 77 K on a micromeritics (BSD instrument, Beijing, China). Morphologies of the photocatalysts were characterized via scanning electron microscopy (SEM, JEOL, Akishima, Japan). To obtain Fourier transform infrared spectra (FTIR), the

photocatalysts were mixed in potassium bromide to form a pellet and characterized on a Thermo Scientific Nicolet iS50 FT-IR spectrometer (Thermo Scientific, Waltham, MA, USA). X-ray photoelectron spectroscopy (XPS) spectra were obtained on X-ray photoelectron spectrometer (Escalab 250Xi, Thermo Scientific, Waltham, MA, USA). With C 1s peak (284.8 eV, surface adventitious carbon) as reference, the binding energy of all photocatalysts was calibrated. The photoluminescence (PL) spectra were analyzed using a FLS980 Series of Fluorescence Spectrometers with an excitation wavelength of 390 nm (OXFORD instrument, Oxford, England). The electrochemical impedance spectroscopy (EIS) spectra, Mott–Schottky, and transient photocurrent (TPC) response were measured via a CHI660E electrochemical workstation (Chenhua, Shanghai, China) in a standard three-electrode configuration. The light harvesting abilities of the photocatalysts were measured using a UV–Vis spectrophotometer (UH4150, Hitachi, Tokyo, Japan).

### 3.6. Photoelectrochemical Characterizations

The photoelectrochemical characterizations were conducted on a CHI660E electrochemical workstation. A three-electrode configuration with a Pt plate as the counter electrode and an Ag/AgCl electrode as the reference electrode was used. The working electrode was prepared on a carbon cloth, which was first cleaned in isopropyl alcohol and naphthyl hydroxide for 40 min, respectively, by ultrasonication. Afterwards, the carbon cloth was dried at 60 °C overnight. Typically, the photocatalyst (10 mg) was ultrasonicated in isopropyl alcohol (1.0 mL) and naphthyl hydroxide (30 μL) to form a homogeneous slurry, then the slurry was dispersed onto the carbon cloth with an area of 1.0 cm$^2$ and dried. The electrolyte was then applied for Mott–Schottky, photocurrent, and electrochemical impedance spectra (EIS) measurement in 0.1 M $Na_2SO_4$ solution.

### 3.7. Photocatalytic Activity

The photoreduction of Cr(VI) by $Ti_3C_2$, UiO-66-$NH_2$, and $Ti_3C_2$/UiO-66-$NH_2$ composites was conducted at room temperature under visible light irradiation. A 300 W Xe lamp (420 nm < λ < 780 nm) was utilized as the light source, a home-made photoreactor as the photoreaction vessel, and 10 mg of photocatalyst was added to a 40 mL $K_2Cr_2O_7$ solution with a concentration of 100 mg/L and pH of 2. The suspension solution was first stirred in the dark for 30 min, and then 2 mL of the suspension was collected every 10 min after irradiation using a 300 W Xe light with a L42 light cutter for 40 min. The obtained suspension solution was filtered with a 0.22 μm microporous membrane, and analyzed via the diphenylcarbazide colorimetric method, using a UV–Vis Spectro photometer (U-3900, Hitachi, Japan). The absorbance at 542 nm (the maximum absorption wavelength of Cr(VI)) gradually decreased as reaction time increased [28,29]. The removal efficiency of Cr(VI) was calculated using the following equation: Cr(VI) removal (%) = (1 − C/C$_0$) × 100%. For the recycling experiment, 10 mg of 2-T/U was added into 40 mL of $K_2Cr_2O_7$ solution (100 mg/L, pH = 2) stirring for 30 min in dark. Then, the suspension was irradiated using a 300 W Xe light with a L42 light cutter for 40 min. After the reaction, 2 mL of the suspension was collected and filtered with a 0.22 μm microporous membrane, and analyzed via the diphenylcarbazide colorimetric method, using a UV–Vis Spectro photometer (U-3900, Hitachi, Japan). The suspension was then centrifuged at 10,000 rpm for 5 min and washed with water until the pH reach 6. Then the obtained 2-T/U was dried at 60 °C and used for the next run. It should be noted that a little of the samples (around 2 mg) was lost during operation, and a little fresh sample was added to make sure the weight of 2-T/U was maintained at 10 mg for the next run.

### 3.8. Standard Curve Diagram

First, 0.25 g potassium dichromate, deionized water, and hydrochloric acid solution were added to a 250 mL volumetric flask to prepare a mother potassium dichromate solution with a pH = 2 and a concentration of 1000 mg/L, respectively. Using the mother solution, potassium dichromate solutions of 0, 20, 40, 60, 80, and 100 ppm with pH = 2

were prepared. The potassium dichromate solutions with different concentrations were measured using a UV–Vis Spectrophotometer (U-3900, Hitachi, Japan), using the diphenyl-carbazide colorimetric method (Figure S7).

## 4. Conclusions

In this work, we synthesized several $Ti_3C_2$/UiO-66-NH$_2$ composites using an in situ solvothermal method. The ratio of $Ti_3C_2$ to UiO-66-NH$_2$ can be easily adjusted. Contents of 5 wt%, 10 wt%, 15 wt%, and 20 wt% $Ti_3C_2$ in $Ti_3C_2$/UiO-66-NH$_2$ composites were obtained by adjusting the ratio of $Ti_3C_2$ and the precursor UiO-66-NH$_2$, and named as 1-T/U, 2-T/U, 3-T/U, and 4-T/U, respectively. The successful synthesis of $Ti_3C_2$/UiO-66-NH$_2$ composites was proven by PXRD, SEM, XPS, FT-IR, etc. The morphology of both UiO-66-NH$_2$ and $Ti_3C_2$ were maintained in the $Ti_3C_2$/UiO-66-NH$_2$ composites, and the UiO-66-NH$_2$ nanoparticles were dispersed on the surface and slits of $Ti_3C$. A strong interfacial interaction between $Ti_3C_2$ and UiO-66-NH$_2$ was formed, as indicated by the XPS. The optimized $Ti_3C_2$/UiO-66-NH$_2$ with 10 wt% $Ti_3C_2$, named as 2-T/U, showed the best photoreduction performance, able to remove 100% $Cr_2O_7^{2-}$ in 40 min. The photoreduction kinetics of $Cr_2O_7^{2-}$ by 2-T/U were also studied, with a photoreduction rate of 0.0871 min$^{-1}$, which is about 2.6 times higher than that of the pure UiO-66-NH. The much higher photoreduction rate of 2-T/U might be ascribed to the formed heterojunction structure between $Ti_3C_2$ and UiO-66-NH$_2$, which can promote the separation of photogenerated charge carriers. This research provides a new type of efficient and environmentally friendly photocatalyst for the photoreduction of $Cr_2O_7^{2-}$.

**Supplementary Materials:** The following supporting information can be downloaded at: https://www.mdpi.com/article/10.3390/catal13050876/s1, Figure S1: EDX (a) and the corresponding area (b) of 4-T/U; Figure S2: Mapping of 4-T/U; Figure S3: XPS spectra of C 1s in 2-T/U and UiO-66-NH$_2$; Figure S4: Digital photographs of all adsorption and photoreduction experiments by 10 mg of 2-T/U. (a) $K_2Cr_2O_7$ solution and (b) diphenylcarbazide colorimetric method treated $K_2Cr_2O_7$ solution. The times for samples from left to right for both (a) and (b) are −30, 0, 10, 20, 30, 40 min; Figure S5: The effect of catalyst dose (a) and substrate concentration (b) on photoreduction performance; Figure S6: PXRD patterns of 2-T/U before and after photocatalysis; Figure S7: Standard curve of potassium dichromate solution; Table S1: Mass and atom percentage of O, N, Ti, F, Zr elements in 4-T/U; Table S2: Comparison of photoreduction performance of our results with other photocatalysts [60–68]; Table S3: Kinetics values of photocatalysts in this work; Table S4: Kinetics values of photocatalysts in previous works [60–68].

**Author Contributions:** Conceptualization, X.W. and J.G.; methodology, H.H. and X.W.; software, H.H. and X.W.; validation, H.H. and X.W.; formal analysis, H.H. and X.W.; investigation, H.H., Q.Y., W.W., X.F., X.R. and D.K.; resources, X.W. and J.G.; data curation, X.W. and J.G.; writing—original draft preparation, H.H.; writing—review and editing, X.W.; visualization, H.H.; supervision, X.W. and J.G.; project administration, X.W. and J.G.; funding acquisition, X.W. and J.G. All authors have read and agreed to the published version of the manuscript.

**Funding:** We are grateful for the financial support from the Guangdong Basic and Applied Basic Research Foundation (No. 2020A1515110003), the National Natural Science Foundation of China (No. 22001094), research funds of Zhejiang Sci-Tech University (No. 21212302-Y), and the open fund of Guangdong Provincial Key Laboratory of Functional Supramolecular Coordination Materials and Applications (No. 2022A06).

**Data Availability Statement:** Not applicable.

**Conflicts of Interest:** The authors declare no conflict of interest.

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
