# Peer review of "In Situ Growth of Ti3C2/UiO-66-NH2 Composites for Photoreduction of Cr(VI)"

_catalysts, doi:10.3390/catal13050876_

Round 1

Reviewer 1 Report

Huan He wrote this manuscript and reported on the Ti3C2UiO-66-NH2 composites for enhanced photocatalytic removal of Cr(VI). From my perspective, this manuscript contains information that can interest the scientific community, and I recommend its publication. However, amendments must be made before the final publication. Below are listed my observations.

1. Please label all the XRD peaks in Fig. 1a.

2. Please provides the TEM, HRTEM, and EDS mapping to analyze the Ti3C2UiO-66-NH2 composites in the revised manuscript or supporting information.

3. Recycling experiments should be provided, and XRD or XPS should characterize Ti3C2UiO-66-NH2 composites after photocatalytic reaction to check the stability of the catalyst.

4. In this study, the surface area is a significant parameter. Therefore, I strongly suggest adding the BET surface area measurement for all samples in the revised manuscript or supporting information.

5. The necessary experiments, such as the trapping experiments for hydroxide radical and the ESR experiments for the O2· should further explore the photocatalytic reaction mechanism. In addition, some papers could be referenced to improve the manuscript further (e. g., Applied Catalysis B: Environmental 263 (2020)11730. Applied Catalysis B: Environmental 260 (2020) 118181. Applied Surface Science 498 (2019) 143850.).

The authors need to re-check this manuscript for spelling and grammatical errors.

Reviewer 2 Report

I reviewed your manuscript “In-situ Growth of Ti3C2/UiO-66-NH2 Composites for Enhanced 2 Photocatalytic Removal of Cr(VI).” very judiciously. The work carried out in the manuscript is very interesting and seems scientifically logical. The authors have added good technical value and knowledge to remove the  Cr(VI) However, there are numerous inaccuracies in this work and before publication, it needs to restructure the research manuscript properly and diligently as the current presentation is not acceptable. Therefore, I would like to recommend this article for "Major Revision".

1.    Title

The title is too long. It should be short, meaningful and attractive. Please modify the title.

2.    Introduction

The novelty of this work was not specified; authors should discuss the novelty of their work in the introduction section. The author should made comparison between their photocatalysts and already reported photocatalysts like hexaferrites materials. The author should also read the research articles related to the wastewater purification, chemical physics letters 805, (2022), 139939, 431–440, New J. Chem., 2022,46, 19848 and doi.org/10.1080/03067319.2022.2032014

3.    Materials and methods

This section should be before the structural analysis. The percentage purity of all the chemicals utilized must be reported. The authors should add the schematic diagram to show the mechanism of Mexene samples synthesis. See Journal of Materials Science: Materials in Electronics 32(7104):1-14 for this purpose. Moreover, photodegradation experiment should also in the material and method section.

XRD analysis

Author should give the JCPDS no while discussing the XRD. Only reference is not enough. Give reason of disappearance of peaks in TiAlC.

SEM& EDX

 There is no connection between SEM and XRD analysis. EDX analysis must be added in the manuscript for atomic as well as weight %. Photodegradation

FTIR

The author should explain the non-appearance of any peaks of Ti3C2 even in fingerprint region ( just a flat line ) ?

Band gap

The line intercepts in band gap determination are wrongly drawn and not exist in the frame along x axis.  The author provide the band gap of only two samples. Please add band gap of all the samples and made comparison which sample have good band gap for photodegradation experiments.

The author reported the BET analysis that is performed to check the adsorption character of the catalyst. But, I see no adsorption experiments here in manuscript. I recommend to perform adsorption analysis and then made comparison between adsorption and photocatalysis. Moreover, the effect of pH, catalyst dose, substrate concentration and effect of temperature is missing.

I also strongly recommend to provide XRD analysis of the samples after photodegradation experiments because photocatalytic properties of the synthesized samples are highly sensitive to the purity of the samples.

4.    Comparison should be made between already reported and author’s submitted article.

5.    I suggest the author to add the digital Photographs of all adsorption and degradation experiments (before degradation and after degradation photograph).

6.    Lot of literature available where the degradation and adsorption of dyes has been reported in similar duration, therefore the novelty ion this work is missing. The author should explain the novelty in proper way.

7.     The author should add the table containing kinetics values like R2, rate constant and % degradation. Another table should be inserted for compassion among reported and this work.  

8.    Revise the conclusion section and add meaningful and numerical values to make it more attractive and easy understanding for new researchers.

Reviewer 3 Report

Dear Editor,

Thanks for inviting me for review the manuscript entitled In-situ Growth of Ti3C2/UiO-66-NH2 Composites for Enhanced Photocatalytic Removal of Cr(VI) authored by Huan He, Junkuo Gao, etc.

The work reports synthesis of Ti3C2/UiO-66-NH2 heterojunction catalyst for photocatalytic reduction of Cr(VI).

The crystal structure, morphology, optical and electronic properties were characterized to demonstrate the merit of the heterojunction. The performance of photocatalytic reduction of Cr(VI) have also been tested. However, there are still some issues to be addressed to make the manuscript more complete before accepting by Catalysts.

1)      Please explain disappearance of the main peak of XRD for Ti3C2 in the composite materials.

2)      As can be seen from Fig. 2c, the Ti3C2 stacked sheets are covered by multi-layered MOF crystals, compared to the degradation profile of Fig. 7a that shows a adsorptive/photocatalytic removal of Cr(VI) of around 80%, please estimate the contribution of the heterojunction surface to the final removal of Cr(VI), besides the adsorptive removal and photocatalytic removal of Cr(VI) by MOFs.

3)      Is there any evidence showing O2 production in the solution, as the author stated the holes can oxidize H2O to O2 (Line 242)?

4)      Please explain how the catalyst recyclability experiments were carried out.

5)      Minor revisions:

Line 48, reference 15 is not suitable to be cited here as it talks the N2 fixation by TiO2, it is recommended to cite some up-to-date review articles here so that readers can gain a knowledgeable background in the field of Cr(VI) reduction by photocatalysis relevant to MOFs and layered catalysts, for example, Chemosphere 303 (2022) 134949; Materials Research Bulletin 147(2022)111636, etc.

Line 266, font of “oC”

Line 288, “is50” change to “is 50”.

Line 295, please provide the brand name of the machine instead of which group.

The English is well to understand, while some spelling and format errors should be double-checked.

Round 2

Reviewer 1 Report

I would like to recommend this paper for publication, as it meets the publication criterion.

English language fine.

Reviewer 2 Report

Accept

Minor editing required